# Positive cardiac inotrope omecamtiv mecarbil activates muscle despite suppressing the myosin working stroke

Michael S. Woody [1], Michael J. Greenberg [2,4], Bipasha Barua[3], Donald A. Winkelmann[3], Yale E. Goldman[2] & E. Michael Ostap[2]

Omecamtiv mecarbil (OM) is a positive cardiac inotrope in phase-3 clinical trials for treatment of heart failure. Although initially described as a direct myosin activator, subsequent studies are at odds with this description and do not explain OM-mediated increases in cardiac performance. Here we show, via single-molecule, biophysical experiments on cardiac myosin, that OM suppresses myosin's working stroke and prolongs actomyosin attachment 5-fold, which explains inhibitory actions of the drug observed in vitro. OM also causes the actin-detachment rate to become independent of both applied load and ATP concentration. Surprisingly, increased myocardial force output in the presence of OM can be explained by cooperative thin-filament activation by OM-inhibited myosin molecules. Selective suppression of myosin is an unanticipated route to muscle activation that may guide future development of therapeutic drugs.

[1] Graduate Group in Biochemistry and Molecular Biophysics, Perelman School of Medicine, University of Pennsylvania, 700A Clinical Research Building, Philadelphia, PA 19104-6085, USA. [2] Pennsylvania Muscle Institute, Perelman School of Medicine, University of Pennsylvania, 700A Clinical Research Building, Philadelphia, PA 19104-6085, USA. [3] Department of Pathology and Laboratory Medicine, Robert Wood Johnson Medical School, Rutgers University, 675 Hoes Lane, Piscataway, NJ 08854, USA. [4] Present address: Department of Biochemistry and Molecular Biophysics, Washington University in Saint Louis, St. Louis 63110 MO, USA. Correspondence and requests for materials should be addressed to Y.E.G. (email: goldmany@pennmedicine.upenn.edu) or to E.M.O. (email: ostap@pennmedicine.upenn.edu)

Omecamtiv mecarbil (OM) is a positive cardiac inotropic agent that increases cardiac performance in failing hearts, as measured by left ventricular fractional shortening and ejection fraction, in both animal models[1] and humans[2–4]. OM was initially described as a direct binder and activator of β-cardiac myosin (MYH7), the molecular motor responsible for powering contraction[1]. The drug has generated great excitement, since a pharmaceutical that directly improves myosin activity holds the promise of avoiding side effects that can occur with drugs that target calcium signaling or other upstream regulators of contraction[5].

Myosin activation by OM was proposed to be the result of an increase in the actin-activated rate of phosphate release (Fig. 1, step 5) without changing the rate of ADP release (Fig. 1, step 6)[1]. This kinetic modification is expected to increase the fraction of the ATPase cycle in which myosins are bound to actin in a force-bearing state (i.e., increasing the duty ratio), resulting in higher force production without affecting the rate of shortening[1]. Subsequent studies confirmed that OM increases the rate of phosphate release[6,7]; however, it was also shown that OM inhibits the velocity of actin gliding in the in vitro motility assay at all concentrations tested[6,8–10] and reduces the rate of tension development and relaxation in myocytes at micromolar concentrations[1,9,11–13]. These phenomena, along with observations of decreased isometric force in fully activated cardiomyocytes[11,14], are inconsistent with the originally proposed model for OM-activation of myosin in muscle.

To better understand the molecular mechanisms by which OM increases contractility, we utilize single-molecule optical trapping to directly measure the effects of OM on the working stroke size and attachment lifetime of individual recombinant-expressed, human β-cardiac myosin molecules. We find that OM suppresses the myosin working stroke and prolongs the actin-attachment lifetime at physiological ATP concentrations and at therapeutic OM concentrations. Simulations of the ensemble behavior of myosins in the presence of OM provide information on the molecular mechanism of OM and how this drug can increase force production in hearts, while also inhibiting cardiomyocyte force production under high calcium and/or high OM concentrations and inhibiting in vitro actin gliding velocity. The results also point to a mode of muscle activation by the selective modulation of a sub-population of myosin molecules.

## Results

**OM reduces the size of myosin's working stroke**. We kinetically and mechanically characterized the interaction of a recombinant, human β-cardiac heavy-meromyosin (HMM) construct with actin in the presence and absence of OM using single-molecule, optical trapping. In this assay, a single actin filament suspended between two optically trapped polystyrene beads (known as an actin dumbbell) is brought into proximity with a single myosin molecule adsorbed to a pedestal bead, which is anchored to the coverslip surface[15,16]. Single myosin molecules bind and displace the dumbbell when it is near the pedestal, and binding events are detected by analysis of the covariance of the bead position fluctuations[17,18] (see Methods). Binding events as short as 16 ms can be detected, and the displacement of the dumbbell by the myosin working stroke is determined with sub-nanometer resolution (Fig. 1, inset).

In the absence of OM, we observed directional displacements of the actin filament consistent with previous measurements (Fig. 2a, b). Full length porcine β-cardiac myosin has been shown to have a working stroke composed of two sub-steps, where an initial displacement associated with the release of phosphate (4.7 nm) is followed by a second displacement associated with ADP

release (1.9 nm)[19]. By averaging ensembles of single-molecule interactions aligned at their beginnings and ends, we also resolved a two-step working stroke for human β-cardiac myosin at low MgATP concentrations (200 nM MgATP, Supplementary Fig. 1). The initial displacement of $4.2 \pm 0.4$ nm (standard error of the mean, sample sizes located in Supplementary Table 1) is followed by a second displacement of $1.5 \pm 0.3$ nm, resulting in a $5.7 \pm 0.3$ nm total working stroke (Supplementary Fig. 1). At a near-physiological MgATP concentration of 4 mM we observe a total working stroke of $5.4 \pm 0.2$ nm, but we do not fully resolve the second displacement due to the rapid binding of ATP and subsequent detachment of myosin ($\sim$500 s$^{-1}$)[6] immediately following the second displacement.

In the presence of a high OM concentration (10 μM), myosin attachments to actin were clearly resolved via a decrease in the covariance signal (Fig. 2c, d). Strikingly, however, there was no clearly discernable working stroke observed (Fig. 2e). The average observed stroke size was $0.4 \pm 0.2$ nm, and never exceeded 0.75 nm for any individual myosin molecule. The optical trap stiffness (0.06–0.08 pN nm$^{-1}$) was much lower than the stiffness of the myosin (0.5–2 pN nm$^{-1}$ [20–22]) and nearly identical in the absence and presence of OM, so it is unlikely that the working stoke was suppressed by the small mechanical resistance imposed by the trap ($\sim$0.5 pN for a 5.5 nm displacement).

Over a range of OM concentrations (0 nM, 50 nM, 100 nM, 200 nM, 500 nM, 10 μM), the calculated average size of the working strokes decreased from $5.4 \pm 0.2$ nm to $0.4 \pm 0.2$ nm in an OM concentration-dependent manner, with an EC$_{50}$ of $101 \pm 25$ nM (Fig. 2f). This EC$_{50}$ is near the clinically relevant plasma concentration of OM of 100–600 nM[23].

We hypothesize that when OM is bound to myosin it completely inhibits the working stroke, and the probability of OM being bound to myosin during an interaction varies with OM concentration (e.g., at the EC$_{50}$ concentration, half of the observed actomyosin interactions occur with OM bound to myosin and have a suppressed working stroke, while the other half of interactions exhibit a full working stroke). We tested this hypothesis by assuming a model where the distributions of the individual stroke sizes were described by the linear combination of two Gaussian distributions, each with its own mean stroke size. We performed a MLE global fit across all observed OM concentrations in the software MEMLET[24] to determine whether the best-fit parameters from this model were consistent with our hypothesis (see Methods). The stroke sizes and widths of the two distributions were shared across all OM concentrations, while the relative proportion of each population could vary. The best-fit parameters to this model gave two populations with working strokes of 5.46 nm and 0.18 nm. The fraction of events with a mean working stroke of 5.46 nm decreased from 100% to 8.8% as the OM concentration increased from 0 to 10 μM with an apparent EC$_{50}$ of 87.5 nM $\pm$ 31 nM (Supplementary Figs. 2 and 3, Supplementary Table 2) which agrees with the EC$_{50}$ for the average step size of $101 \pm 25$ nM. This result is consistent with a model in which myosin performs it full working stroke when OM is not bound but has virtually no net displacement when OM is bound, and the proportion of interactions that occur with OM bound depends on the OM concentration.

**OM prolongs myosin attachment durations at physiological ATP**. Actin-bound durations were well-described by single-exponential functions at all MgATP concentrations studied in the absence of OM, as indicated by the log-likelihood ratio test performed in MEMLET[24] (Supplementary Figs. 4, 5 and Supplementary Table 3). The rates of detachment were linearly related to [MgATP] at low concentrations (0.2–10 μM, Fig. 3e), yielding

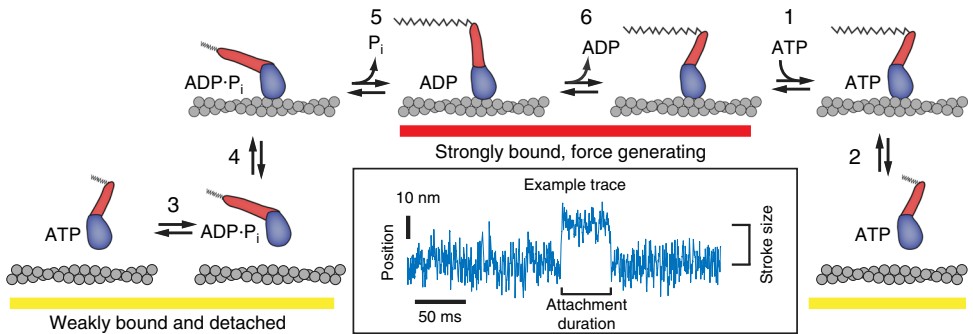

**Fig. 1** Biochemical cycle of cardiac myosin. Omecamtiv mecarbil has been shown to increase the rate of phosphate release (step 5) and bias the ATP hydrolysis step (step 3) towards the post-hydrolysis M·ADP·P$_i$ state, which is proposed to cause myosin to enter the strong binding states (red underline) more rapidly. Other biochemical steps have been previously shown through stop-flow biochemical experiments to be nearly unchanged by the presence of OM. Inset: Example optical trapping trace of the position (median filtered with 0.4 ms window) of an actin filament during one interaction with a single-myosin molecule reproduced from Fig. 2b. The step size and attachment duration of these interactions can be measured as shown

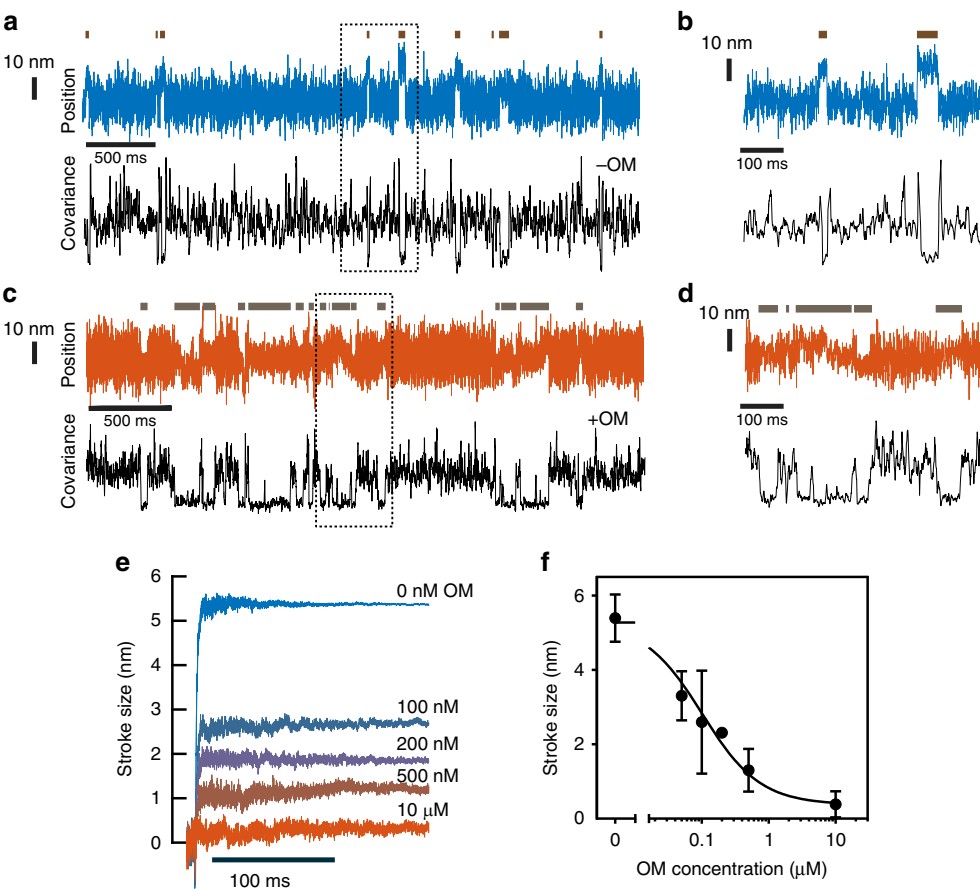

**Fig. 2** The effect of OM on myosin working stroke size. **a** Example trace of the position of one bead during several interactions of cardiac myosin with actin in the absence of OM (blue). The covariance of the two beads' positions is shown in black and was used to determine when a binding event occurred, as indicated by the dark horizontal lines above the position trace. **b** An expanded section of the data inside the dashed box in **a**, where two clear interactions can be visualized. **c, d** Example trace similar to that in **a** and **b**, but with 10 μM OM present. The interactions are more difficult to distinguish in the position trace (red) but are clear from the covariance (black). Position traces in **a–d** are median filtered with a 0.4 ms window. **e** Binding events were synchronized at their starts and averaged forward in time to show the average stroke size observed in the presence of OM ranging from 0 to 10 μM. Average stroke size decreases with increasing OM concentration. **f** The average observed stroke size was decreased by OM in a dose-dependent manner. Error bars give the standard deviation of the mean step sizes from each molecule observed. *N*-values are presented in Supplementary Table 1

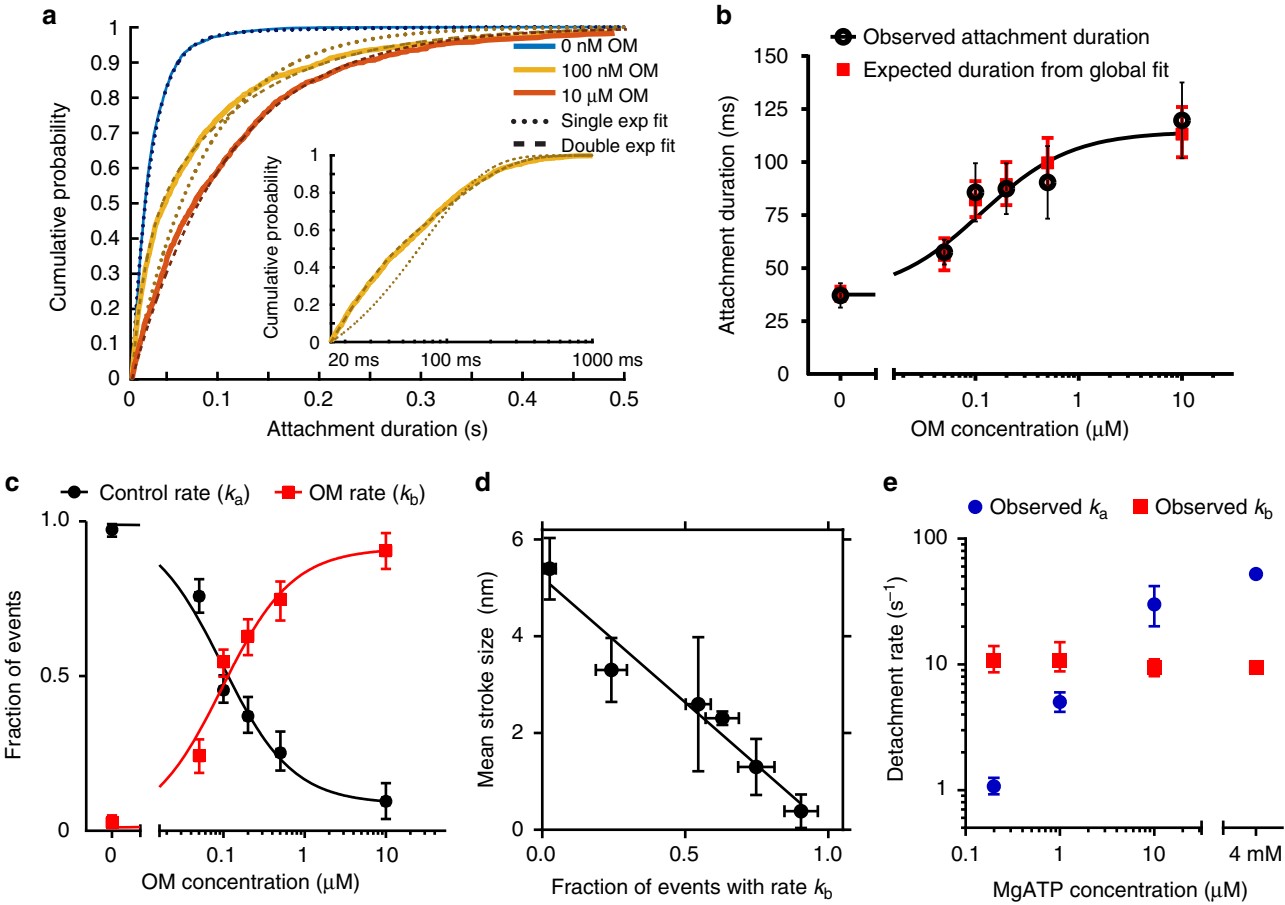

**Fig. 3** Actin-attachment durations as a function of OM concentration. **a** Cumulative distributions of the actomyosin attachment durations (solid lines) at 4 mM MgATP. Without OM, the attachment durations are well-described by a single-exponential distribution (dotted blue line); however, at 100 nM and 10 μM OM, double exponential distributions were required (yellow and red dashed lines). Inset: 100 nM OM durations on logarithmic x-scale, highlighting the two phases of detachment. **b** Concentration-dependent effect of OM prolonging the mean observed attachment duration (black) at 4 mM MgATP. Black error bars show the standard deviation of the mean durations from each molecule. Red squares show the expected duration calculated from the global fit to durations. **c** The fraction of events which were found to detach at $k_a$, (black), or at the OM-associated rate ($k_b$, red) from the durations global fit as a function of OM concentration. **d** Observed step size linearly correlates with the fraction of events which detach at the OM-associated rate, $k_b$. Vertical error bars are the standard deviation of the mean step from each molecule studied. **e** Detachment rates at 10 μM OM as a function of MgATP concentration. Rate $k_a$ (blue) was proportional to MgATP concentration at low ATP concentrations. Rate $k_b$ (red) was only observed in the presence of the drug and was independent of the ATP concentration at all concentrations studied. Unless otherwise noted, error bars from all panels show the 95% confidence intervals obtained via bootstrapping. N-values are presented in Supplementary Table 1

an apparent second-order rate constant for MgATP binding and detachment ($3.0 \pm 0.14\ \mu M^{-1}\ s^{-1}$), which is consistent with biochemical measurements[6]. At saturating ATP (4 mM MgATP), the mean detachment rate of $47.7\ s^{-1}$ ($43.2$–$51.9\ s^{-1}$ 95% CI, Fig. 3a) is consistent with ADP release limiting the rate of actin detachment as predicted by biochemical experiments and shown previously[6,19,25].

High concentrations of OM (10 μM) decreased the actin-detachment rate 5-fold to $9.4\ s^{-1}$ ($8.6$–$10.3\ s^{-1}$ 95% CI), as calculated by a single-exponential fit. As the OM concentration varied from 0 to 10 μM, the mean observed attachment duration was prolonged with an $EC_{50}$ of $114 \pm 38$ nM (Fig. 3b, Supplementary Note 1).

The durations of attachment with OM present were not well-described by single-exponential distributions but were best-fit by two-component exponential functions as determined by log-likelihood ratio testing[24] (Fig. 3a, inset, Supplementary Fig. 4 Supplementary Table 3). At saturating ATP and 50 nM to 500 nM OM, the rate of the faster phase ($k_a$) is similar to the detachment rate of myosin in the absence of OM and the

rate of the slow phase ($k_b$) is similar to the detachment rate in the presence of 10 μM OM (Supplementary Fig. 4, Supplementary Table 3). A global fit of a double exponential distribution was performed for the 0–10 μM OM datasets simultaneously with rates $k_a$ and $k_b$ being shared between the datasets. Only the relative proportion of events which dissociated at each rate was allowed to vary between OM concentrations[24]. This procedure yielded two common rates similar to those observed at the 0 μM and 10 μM OM concentrations ($k_a = 52\ s^{-1}$ and $k_b = 9.4\ s^{-1}$), and the relative fraction of events dissociating at rate $k_b$ increased monotonically with increasing OM concentration with an $EC_{50}$ of $114 \pm 28$ nM (Fig. 3c). This supports the idea that the concentration of OM affects the likelihood that myosin will be bound to OM when it binds to actin, and that when OM is bound, the rate of detachment of myosin is changed from $k_a$ to $k_b$.

The proportions of events detaching at the OM-associated rate, $k_b$, show a linear relationship with the observed working stroke sizes (Fig. 3d). The observed stroke size decreased as more events detached at rate $k_b$. This relationship supports a model in which

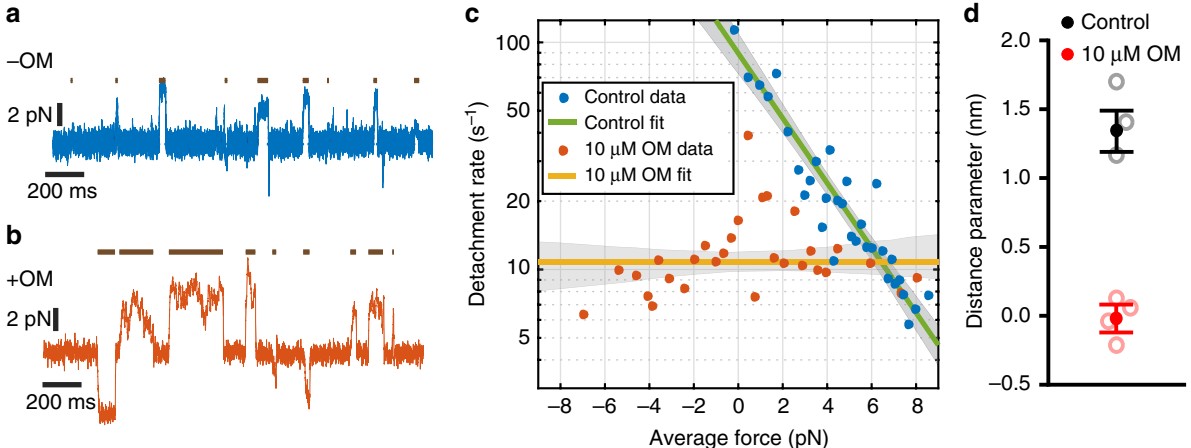

**Fig. 4** Force dependence of detachment is reduced by OM. **a**, **b** Example traces of force on the motor bead (blue, red) with the trap's isometric feedback system engaged. Events, as detected by covariance, are indicated by dark lines above the force traces. **a** In the absence of OM (blue), forces are predominantly in the positive direction (myosin is under resisting load). **b** In the presence of 10 μM OM (red), forces are generated in both directions. **c** The observed detachment rates in the absence of OM (blue) and at 10 μM OM (red) as a function of applied load are shown as circles at the average force and rate of 20 events sorted by force. Green and yellow lines show the detachment rate calculated from Eq. 1 and parameters from the MLE fit in the absence of and presence of 10 μM OM, respectively, with 95% confidence intervals shown as shaded gray areas. Data are composed of 665 and 583 binding events from 3 and 4 molecules for the absence of OM and 10 μM OM, respectively. **d** Distance parameter estimates for the control (no OM) and 10 μM OM data from all observed molecules (closed circles) are shown with 95% confidence intervals from bootstrapping and the estimated distance parameters from individual molecules (open circles)

both the stroke size is suppressed and the detachment rate is changed when OM is bound to myosin.

**Detachment of OM-bound myosin is independent of ATP binding**. At low ATP concentrations (≤10 μM), where the acto-myosin dissociation rate is dependent on the ATP concentration, the kinetics of actomyosin dissociation with OM present were again best described by two-component exponential distributions (Supplementary Fig. 5). One of the observed rates ($k_b$) was near $10\,\text{s}^{-1}$ for all MgATP concentrations studied (Fig. 3e, red squares). The other rate ($k_a$) varied with ATP concentration and was consistent with the previously measured second-order rate constant of $3\,\mu\text{M}^{-1}\,\text{s}^{-1}$ for ATP binding to actomyosin[6]. At MgATP concentrations ≤1 μM MgATP, the ATP-dependent rate of detachment was slower than the OM-associated $10\,\text{s}^{-1}$ detachment rate, while at MgATP concentrations ≥ 10 μM, ATP-induced dissociation was faster than $10\,\text{s}^{-1}$ (Fig. 3e). Detachments with rates $k_a$ and $k_b$ were consistently observed under the same conditions in a single experiment from the same molecule. This result indicates that after OM-bound M·ADP·$P_i$ binds to actin and enters a strongly bound state (step 5; Fig. 1), its subsequent actin dissociation is not through ATP binding to AM (step 1; Fig. 1). Additionally, the dissociation at rate $k_b$ is not due to OM-induced dissociation of the M or M·ADP states from actin, as stopped-flow biochemical experiments indicate that these dissociation rates are substantially slower than $k_b$ and not affected by OM (Supplementary Note 2, Supplementary Fig. 6, Supplementary Table 4). Taken together these results indicate that cycling, OM-bound myosin does not pass through the canonical rigor state that requires ATP binding for detachment, but rather follows a non-canonical pathway.

**Force dependence of detachment rate is reduced by OM**. To probe the force dependence of the attachment lifetime of cardiac myosin, we used an isometric feedback technique[18] previously used to determine the force dependence of various myosin isoforms[16,19,26]. This technique applies a load onto the myosin

molecule by varying the position of one of the optical traps to maintain a constant position of the bead in the opposite trap. This acts to maintain the actin filament (which is inside the feedback loop) in a nearly constant position, allowing myosin to exert an isometric force while it interacts with actin. The value of the applied load is determined by the distribution of myosin binding positions along the actin and the magnitude of the working stroke. In the absence of OM, myosin was able to maintain isometric forces ranging from −1 pN to +7 pN (Fig. 4a), while in the presence of 10 μM OM, the observed forces ranged between −8 pN to +8 pN (Fig. 4b). There was no apparent bias to the direction of the applied forces with OM present, even within a single molecule's interactions with a single actin filament. The force distribution in the presence of OM is likely due to stochastic binding along the actin filament due to Brownian motion unbiased by a working stroke, supporting the result given earlier (Fig. 2) that OM suppresses the working stroke.

The force dependence of the actin-detachment rate can be quantified by fitting the Bell equation[27] to the attachment durations without binning using maximum-likelihood estimation:[24]

$$k(\mathbf{F}) = k_0 e^{\left(\frac{-\mathbf{F}\cdot\mathbf{d}}{k_B T}\right)},\tag{1}$$

where $k_0$ is the rate of detachment in the absence of applied load, **F** is the applied force, **d** is the distance to the force-dependent transition state, and a measure of the degree of force dependence, $k_B$ is Boltzmann's constant, and $T$ is temperature. Binned detachment rates as a function of force and the results of fitting Eq. 1 to the data are shown in Fig. 4c. In the absence of OM, we found that the distance parameter (**d**) was 1.3 nm (1.19–1.49 nm 95% CI), with $k_0 = 89\,\text{s}^{-1}$ (72.9–109 $\text{s}^{-1}$ 95% CI), consistent with previous measurements for β-cardiac myosin[19,25,28]. However, in the presence of 10 μM OM, there was no appreciable change in the detachment rate as a function of load (Fig. 4d). A log-likelihood-ratio statistical test[24] also indicates that fitting the Bell model to the data is not justified over a force-independent single-

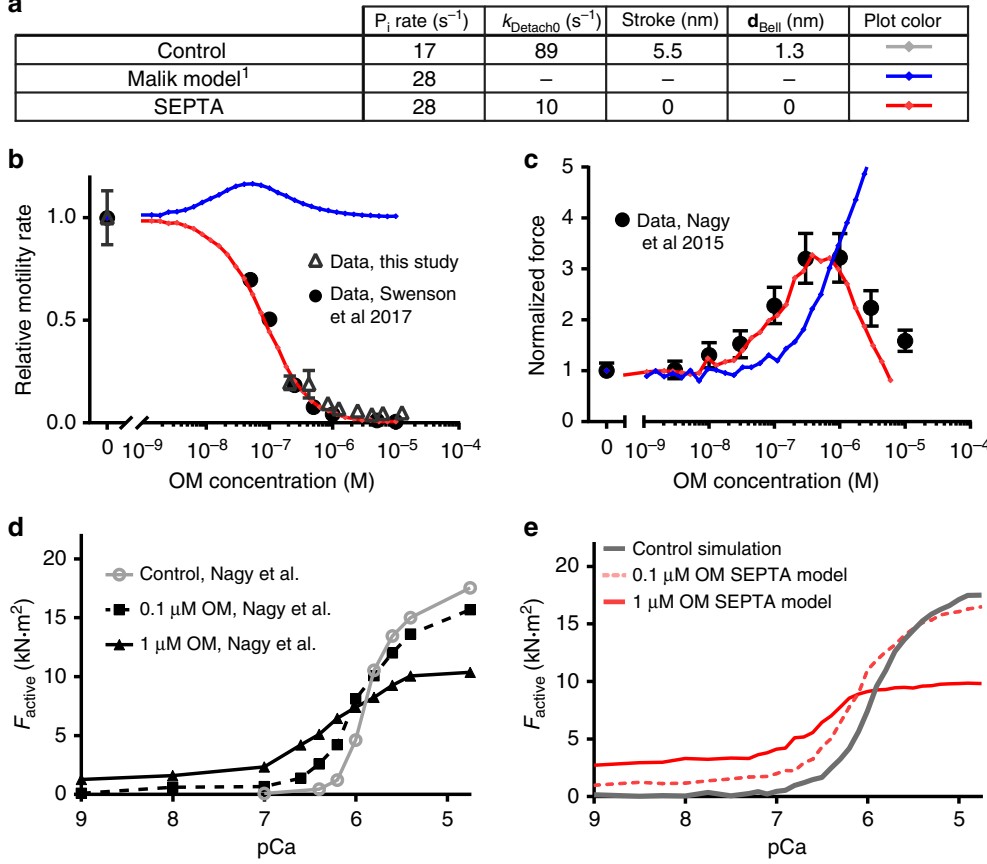

| | $P_i$ rate (s⁻¹) | $k_{Detach0}$ (s⁻¹) | Stroke (nm) | $d_{Bell}$ (nm) | Plot color |
|---|---|---|---|---|---|
| Control | 17 | 89 | 5.5 | 1.3 | |
| Malik model[1] | 28 | – | – | – | |
| SEPTA | 28 | 10 | 0 | 0 | |

**Fig. 5** Simulations of OM's effect on in vitro actin gliding velocity and isometric force of muscle preparations. **a** Summary of parameters used in models in **b**−**e**. **b** Comparison of gliding filament velocity for the protein used in this study (open triangles) and from Swenson et al.[9] (closed circles) to simulated data from the various model parameter sets (colored lines as in **a**). The Stroke Eliminated, Prolonged Time of Attachment (SEPTA) Model (red), with parameters from the single-molecule measurements, fully accounts for the observed marked decrease in vitro velocity as a function of OM concentration. Motility error bars are standard deviations of velocities from individual filaments (n of filaments = 92–6709). **c** Comparison of simulated, normalized, isometric forces at an intermediate calcium concentration (15% activation, lines colored as in **a**) and results from Nagy et al.[11] who reported a bell-shaped force response of permeabilized myocardial trabeculae as a function of OM (black circles). The SEPTA model (red) shows a similar biphasic shape. **d** Active force data reproduced from Nagy et al. as a function of pCa ($-\log [Ca^{2+}]$) at 0, 100 nM and 1 μM OM. **e** Simulated isometric force at 0 OM (gray line), and using the SEPTA model at 100 nM (red dashed line) and 1 μM OM (red, solid line) for comparison with the experimental data in **d**. The SEPTA model recapitulates the leftward shift in the pCa-tension curve (calcium sensitization) and decreased force production at fully activating $[Ca^{2+}]$

exponential distribution ($p = 0.62$). This reduction in force dependence and prolonged attachment lifetime by OM is similar to recently published results using an alternative optical trapping method[28].

**Simulation of myosin ensemble behavior in the presence of OM.** We carried out simulations to determine whether the results we observed in the optical trapping assay could explain the effect of OM on the actin gliding rate in the in vitro motility assay[6,8–10] and on the calcium-dependence of contractile function of cardiomyocytes[11,14]. We used a simple, two-state model of the actomyosin cycle along with cooperative activation of the thin filament as simulated by Walcott et al.[29,30] (see Methods). This model was used to test whether the effects of OM on attachment rate, detachment rate, step size, and force dependence would alter gliding velocity and isometric force as a function of OM and calcium. We utilized parameter values from the literature[6,20,30] combined with the detachment rate, stroke size, and force dependence measured in our experiments in the presence and absence of OM (Supplementary Table 5). We tested six possible models of OM's effects (Fig. 5a, Supplementary Fig. 8), including one where only the phosphate release rate was increased (Malik

Model[1]). All other models also included this increase in phosphate release rate, but varied in the size of the stroke, the detachment rate, and/or the force dependence of detachment rate. The model utilizing the parameters measured in this work for step size, force dependence, and attachment duration is referred to as the Stroke Eliminated, Prolonged Time of Attachment (SEPTA) Model.

Simulations of unloaded velocity, using an OM binding affinity of 100 nM, showed that the SEPTA model (Fig. 5b, red) which includes a decreased step size and decreased detachment rate was able to fully account for the drastic reduction in actin gliding velocity we and others[8–10] observe in vitro (Fig. 5b, black circles). It is also possible to simulate a drastic reduction in the motility rate by decreasing the working stroke size without changing the detachment rate (Supplementary Fig. 8, SE model, green). The Malik Model, the previously proposed mechanism of action for OM[1], did not decrease the velocity (Fig. 5b, Malik, blue). It had previously been suggested that the slowed motility could be due to increased force dependence of myosin detachment in the presence of the drug[6,7,9], but a 10-fold increase in the distance parameter, **d**, corresponding to a greatly increased force sensitivity of detachment could not reproduce the observed

suppression of the gliding velocity by OM in the in vitro motility assay (Supplementary Fig. 8, brown).

Experimental measurements using isolated and permeabilized cardiomyocytes have shown a biphasic effect of the OM concentration on isometric force production, with force enhancement peaking at intermediate calcium concentrations[11,14]. The SEPTA model reproduced this biphasic force response in simulations (Fig. 5c, red). Simulations using other parameter variations were able to predict an increase in force production with increasing OM, but only when the working stroke was inhibited did the force decrease at high simulated OM concentrations (Supplementary Fig. 8, SEPTA, SE Models). SEPTA also was the only one of the six combinations tested that reproduced the inhibition of force by OM at high calcium concentrations (pCa 4.75) and the increased calcium sensitivity reported by Nagy et al. and others[9,11,13,14], who demonstrated a leftward shift in the pCa-tension curve (Fig. 5d, e). While the in vitro motility assay simulations used a dissociation constant for OM ($K_d$) of 100 nM, which is consistent with our optical trapping data, in the simulations of myocyte isometric force, $K_d$ for OM was set to 1.2 µM. This value is consistent with the observed effects of the drug in the same type of muscle preparations (permeabilized rat trabeculae) used in Nagy et al.[11] and as discussed in Supplementary Note 3.

## Discussion

OM is not a direct activator of myosin, and its inhibitory effects on motility seem counterintuitive for a drug that improves myocardial function in heart failure. Our single-molecule results provide a revised mechanism of action of this interesting pharmaceutical that explains results from a variety of biochemical, structural, and physiological experiments.

Our optical trapping studies indicate that OM binding to myosin has two main effects. First, it inhibits the size of the working stroke more than 10-fold, from 5.4 nm to less than 0.4 nm. Secondly, actomyosin attachment duration is prolonged at physiological ATP concentrations by 5-fold, and detachment becomes independent of both ATP concentration and force applied to the myosin. The OM effect on both the working stroke and attachment duration occur with an $EC_{50}$ of ~100 nM. A simple model predicts that prolonged attachments lead to increased force production in muscle due to thin-filament activation (see below), while the inhibited step size explains the 30-fold reduction of gliding velocity in vitro and decreased force production in cardiomyocytes observed under either high OM concentrations or at all OM concentrations for fully activating calcium concentrations.

Our discovery of the OM-induced decrease of the myosin working stroke size clarifies the findings of previous studies. Biochemical and structural studies demonstrated that OM stabilizes the pre-power stroke ADP·$P_i$ state[6,31]. Transient-time-resolved distance measurements utilizing fluorescence resonance energy transfer suggest that the rate of a conformational change that correlates with myosin's working stroke is drastically reduced by OM[7], leading to the possibility that myosin detaches from actin before the power stroke occurs. Additionally, molecular dynamics studies of cardiac myosin bound to OM suggest that movement of the myosin lever arm relative to the myosin motor domain may be inhibited by OM[32]. Recently reported in situ studies also have shown evidence that the stroke may be reduced by OM in muscle fibers[33].

Biochemical kinetic experiments have suggested that OM does not affect the actomyosin-detachment rate during cycling[1,6], which is at odds with our findings. However, studies of tension development and relaxation rates of cardiac muscle preparations have indicated that detachment may be slowed by OM[11,12,34]. The previous biochemical conclusions[1,6] were based on the measurement of the rate of ADP release (the step that limits the rate of actin detachment at physiological ATP) from an AM·ADP complex formed by adding ADP to the rigor, AM complex in isolated actomyosin[1,6,9]. Our results indicate that the step that limits actin detachment is not accessible by simply adding ADP, but occurs earlier in the cycle and may not be on the conventional ATPase pathway (Fig. 6). Interestingly, a single-turnover, stopped-flow study showed the rate of ADP release was slowed by OM when myosin proceeded through its cycle from the M·ADP·$P_i$ state[9]. However, the authors of this study concluded that the actin-bound, ADP-isomerization step (AM'·ADP → AM·ADP) found on the canonical ATPase pathway was affected by OM. This interpretation is inconsistent with data presented here (Fig. 3e) since their model would include a normal working stroke displacement and an actin-detachment rate that would be slowed by low MgATP concentrations.

In contrast, our experiments suggest that myosin detaches from actin from an ADP bound or apo state off the typical pathway, since we observe that myosin does not undergo a working stroke while attached to actin. Detachment of cycling myosin from actin without ATP binding has been previously proposed without OM on other grounds[35,36]. Our findings are consistent with the lack of force dependence of actin-detachment rate in the presence of OM (Fig. 4). Forces that resist the myosin working stroke have been shown to slow the ADP isomerization or ADP release step[19,25]. As the working stroke does not occur in the presence of OM, the post-stroke

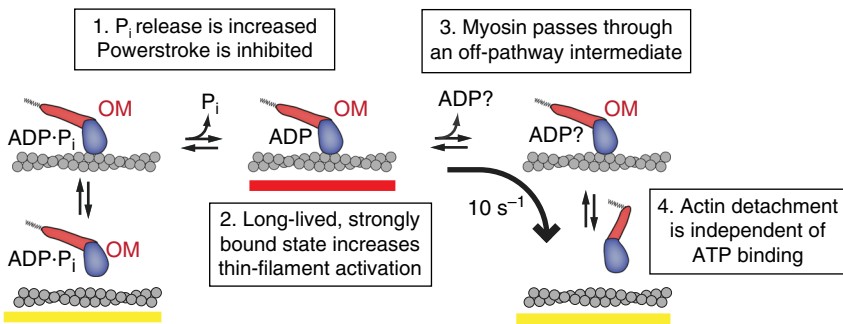

**Fig. 6** Model of OM's effect on cardiac myosin. (1) OM increases the rate of entry into strong binding as previously measured by phosphate release rates, but the force generating power stroke is inhibited. (2) Myosin remains strongly bound to actin (red bar), contributing to increased thin-filament activation at intermediate calcium concentrations. (3) OM disrupts the typical pathway of myosin, causing it to pass through an ADP or apo (nucleotide-free) state with its lever arm still in the pre-power stroke position. (4) Myosin detaches from actin without needing to bind ATP. ATP binding must occur before the cycle can start again

state preceding force-sensitive ADP release is never populated, and the kinetic step that limits detachment is thus not force dependent.

Our SEPTA (Stroke Eliminated, Prolonged Time of Attachment) model reproduced the biphasic effect of isometric force production on OM concentration in cardiomyocytes (Fig. 5). Models that included increased phosphate release rates and/or prolonged actin attachment resulted in monotonic increases in force with increasing OM (Fig. 5c). The force increase is expected, since increasing the phosphate release rate and decreasing the actin-detachment rate lead to a higher duty ratio, with more myosins dwelling in the strongly bound states. However, increasing phosphate release (Fig. 5, Malik Model) and/or prolonging the time attached (Supplementary Fig. 8, PTA Model) alone does not account for the experimentally observed decreases in force at high OM concentrations or at fully activating calcium concentrations (Supplementary Fig. 8c, e). An OM-induced decrease in the working stroke does account for these effects. Although not modeled in our simulations, OM's proposed effect on thick-filament activation also could result in monotonically increasing force production with OM concentration[14]. While thick-filament activation alone cannot explain OM's biphasic effect on force production, it may explain the observed decrease in the Hill coefficient for the isometric force pCa curves which is not reproduced in our simulations.

Our simulations suggest that the experimentally observed increase in cardiomyocyte force production in the presence of sub-micromolar OM concentrations is due to prolonged actin attachment increasing thin-filament activation. OM-bound myosins do not produce a power stroke and cannot generate force, so experimentally observed increases in force must be due to the increased recruitment of drug-free myosin. OM-bound myosin activates the thin-filament regulatory system allowing the cooperative binding of fully functioning myosin[37]. A similar effect on thin-filament activation has previously been shown in low ATP conditions, where the actin-attachment duration of myosin to the thin filament is increased[38,39]. At high concentrations of the drug, when a majority of myosin heads are bound to OM, force is inhibited at all calcium concentrations, since OM-bound myosin does not perform its working stroke. A prediction of the SEPTA model, that is seen experimentally[11,13,14], is that OM does not increase force production at calcium concentrations that fully activate the thin filament, and increasing OM concentrations inhibit force. The maximal force at intermediate calcium (15% activation) is developed when approximately 30% of myosin heads are bound to OM in the conditions of our simulation (Supplementary Figure 7).

The observed effects on working stroke and attachment duration occur at OM concentrations lower than the therapeutic plasma concentration of OM in patients of ~100–600 nM[2,4,23], making these observations relevant to therapeutic applications of the drug (See Supplementary Note 3).

Our single molecule, optical trapping experiments have revealed that at concentrations relevant to therapeutic treatment of patients, OM acts to suppress cardiac myosin's working stroke and to prolong the time of actin attachment. These effects intuitively seem at odds with the positive results of the clinical trials which showed increased cardiac performance in patients treated with the drug. However, our findings can account for the previously unexplained inhibition of the gliding velocity in vitro and force production in cardiomyocytes under saturating calcium and/or high OM concentrations. The previous hypothesis that motility was inhibited by increased force sensitivity of the myosin-detachment rate has been shown to be unlikely, both by direct measurement of the force dependence (Fig. 4) and our modeling results (Supplementary Figure 8). Our simulations

show that the increased attachment lifetime of a small population of non-force generating myosin molecules is sufficient to explain the calcium sensitization effect observed in cardiomyocytes and is the likely explanation for the physiological benefits seen in clinical trials. The concentration of OM in patients has been shown clinically to be crucial, as doses of OM above the therapeutic level slow relaxation to the point where ischemia or other adverse reactions may occur[3]. This knowledge helps to resolve the apparent inconsistencies observed in the wide variety of experiments and applications that have used this promising drug.

This study provides an example of how the perturbation of myosin by a drug can affect more than the kinetic rate constants of its biochemical cycle. To fully characterize a mutation's or drug's effect on myosin, the possibility of off-pathway states being induced by the perturbation should be considered. In addition, the observed modulations of myosin function can cause unexpected functional alterations in the more complex, integrated muscle cell system. Targeted drug development for alleviating functional deficits can benefit from screening for biochemical alterations of myosin's cycle and testing for biophysical effects, and also taking account of the indirect actions that emerge when they are assembled into the full biological environment.

## Methods

**Protein purification, in vitro gliding assays**. An heavy meromyosin (HMM) construct of human β-cardiac myosin (MYH7) was expressed in C2C12 myoblasts and purified[10]. Actin gliding assays were performed using fluorescently labeled actin filaments at 32 °C[10]. OM was obtained from Selleck Chemical (S2623) and a 10 mM stock solution was prepared in DMSO and stored at −80 C.

**Optical trapping assay**. Optical trapping assays were performed at room temperature (20 ± 1 °C). Flow cell chambers were constructed using double-sided tape and vacuum grease[19,40]. The surface of the coverslip was coated with a 0.1% nitrocellulose solution (EMS) mixed with 2.5 μm diameter silica pedestal beads (Polysciences). The main assay buffer (AB) contained 25 mM KCl, 60 mM MOPS, 1 mM DTT, 1 mM MgCl₂, and 1 mM EGTA. Cardiac myosin in myosin buffer (AB with 300 mM KCl) was incubated in the 20–30 μL chamber for 30 s to non-specially adhere to the nitrocellulose coated surface, before being washed out with additional myosin buffer. The concentration of myosin, ranging from 0.02 to 0.1 μg mL⁻¹, was adjusted each day so that only 1 out of 5–10 locations tested showed interaction with the actin filament. The chamber was blocked with 2 incubations of 1 mg mL⁻¹ BSA each lasting 3 min. The experimental solution was added to the flow cell containing MgATP, either 0.1% DMSO for control experiments or OM (in DMSO with a final DMSO concentration of 0.1% for all conditions), 0.1–0.2 nM actin filaments composed of 10% biotin actin (Cytoskeleton) and 90% unlabeled rabbit skeletal actin prepared from cryoground rabbit back muscle (Pel-Freeze)[41] and stabilized by rhodamine-labeled phalloidin (Sigma) and an oxygen scavenging system of ~3 mg mL⁻¹ glucose, and glucose oxidase catalase. For ATP concentrations ≤ 10 μM, the concentration of a freshly made 1 mM MgATP stock solution was verified spectroscopically each day. Polystyrene beads with a diameter of 500 nm (Polysciences) were prepared by incubating ~0.40 ng of beads with 10 μL of 5 mg mL⁻¹ neutravidin solution (Thermo) in water overnight at room temperature before washing four times with AB via centrifugation. Four microliters of a solution containing approximately 5 ng mL⁻¹ neutravidin-coated beads, MgATP, and either 0.1% DMSO or OM was added to the chamber before it was sealed with vacuum grease.

Experiments were performed on a dual-beam optical trapping setup[42] which utilizes a 1064 nm laser for trapping and direct force detection using quadrant photodiodes (JQ-50P, Electro Optical Components Inc.) and a custom-built amplifier. The two beams were produced and controlled by splitting the laser by polarization and passing each beam through a 1-D electro-optical deflector (LTA4-Crystal, Conoptics), which could deflect the beam based on input from a high voltage source (Conoptics, Model 420 Amplifier). Data acquisition, feedback calculations, and beam position control output utilized a LabVIEW Multi-function I/O device with built-in FPGA (PXI-7851) and custom-built virtual instruments (LabVIEW). Data acquisition and digital feedback calculations occurred at 250 kHz.

The microscope utilized a Nikon Plan Apo ×60 water immersion objective (NA 1.2) and a Nikon HNA Oil condenser lens (NA 1.4). One bead was trapped in each of the two 1064 nm beams, with a trap stiffness of 0.06–0.08 pN nm⁻¹, calculated via the power spectrum of the beads' positions[43]. An actin filament 5–10 μm long was attached at each end to the two beads. The position of one bead was adjusted (using a servo-controlled mirror that was positioned conjugate to the back focal of the objective) to stretch the actin filament under 4–6 pN of pretension, reducing the effects of the non-linear compliance of the bead-actin linkage. Pedestal beads were tested by moving the actin filament close and observing whether any

reductions in variance of the measured force signals occurred. When interactions were detected, a piezo-electric stage (Mad City Labs) was used to fine tune the position to maximize the number of interactions per second. A feedback system using an image of the pedestal bead and the nano-positioning stage was used to stabilize the stage position[44,45]. For a given molecule, the position of the stage was slightly shifted along the axis of the filament by 6–12 nm between some data acquisition traces to reduce inhomogeneity of the accessibility of the actin-attachment zones[21].

The isometric feedback experiments[16,18,19,26] were conducted using a digital feedback loop and EODs to steer the beam position. Briefly, a feedback loop held the position of one of the beads (referred to as the transducer) constant by modulating the position of the other trap, known as the motor trap. Since the actin filament is between the two beads and is inside the feedback loop, its position was maintained continuously, allowing the myosin to develop isometric force during its interaction with actin. The excursion of the motor trap was limited to 100–125 nm (corresponding to ~6–10 pN) to prevent entering the non-linear force regime due to the bead being pulled too far from the trap center. The response time of the feedback loop during myosin interactions was ~15–20 ms.

**Optical trap data analysis**. Optical trap data were analyzed by calculating the covariance of the force signal from the two beads with a window of 8–15 ms[16,19,26]. The covariance value from a 30 s recording was fit to a double Gaussian distribution, with the mean of the high covariance value peak representing when myosin was detached and the lower covariance value indicating attachment to actin. Events were selected by determining when the covariance signal passed from the detached value to the attached level and back to the detached level. The time of the binding event start and end were refined by examining the point when covariance crossed a threshold calculated to minimize the overlap between the distributions of bound and unbound covariance values[16,26]. The duration of the event was calculated from these values of the start and end of each event. The total stroke size of the event was found by averaging 1 ms of the actin displacement of both beads 4–10 ms prior to dissociation and subtracting the baseline displacement of the actin position calculated by the average of a 1 ms window 4 ms after the detected end of the event. Depending on experimental conditions (dumbbell length, stiffness of bead-actin linkages, etc) events shorter than 16–30 ms (16 ms for saturating ATP and up to 30 ms for [ATP] < 4 mM) could not be reliably detected and were eliminated from the analysis. This deadtime was determined by taking the size of the covariance window and multiplying it by a factor of 2. Ensemble averaging was performed by aligning the starts of each event and averaging the signals[16,19,26,46] and were weighted such that each molecule contributed equally to the average.

Events from data acquired utilizing isometric feedback were also analyzed by examining the covariance signal, which clearly decreased during actomyosin interactions. While the feedback loop was engaged, we also observed transient (<5 ms) rises in the covariance signal in the middle of interactions that were plainly not associated with detachment but were more likely due to a small conformation change in the myosin which causes both beads to move together simultaneously. These transients in the covariance signal were recorded as detachment events by the initial analysis, but a second pass through the data eliminated these detachments by removing detected unbound events which both lasted less than two times the covariance calculation window and were not associated with a return of the motor bead force to the baseline. The average force during an interaction was calculated by averaging the force on the motor bead starting 2 ms after detected attachment through 2 ms before detachment. The baseline force on the motor bead 4 ms after detachment was subtracted from this average force.

**Attachment duration and step size parameter estimation**. Detachment rates and mean step sizes were calculated using MEMLET[24], a MATLAB-based program that utilized maximum-likelihood estimation to perform parameter estimation without the need for binning. For each set of conditions, parameter estimation was performed on data combined from multiple molecules (estimates from individual molecules yielded similar results to the combined datasets). Because the number of events from each molecule varied due to non-biologically relevant experimental conditions (age of chamber, precise positioning of the actin filament, number of data traces recorded by the user), for each condition, data were weighted such that the data from each molecule counted equally in the parameter estimation process. The multidimensional fitting capability of MEMLET was used to include weights for each duration or step size, and each weight included as an exponential factor acting on the total probability density function (PDF) describing the distribution before the log of the PDF was minimized[24]. The weights were calculated such that the sum of all weights was equal to the number of total points being fit, so that the estimated log-likelihood would be consistent with unweighted fitting results. Duration data were fit to either single exponential or double exponential PDF with weights, while step size data was fit to a weighted double Gaussian distribution in MEMLET[24] (v 1.3). Log-likelihood ratio testing was performed using MEMLET to test for the significance of a two-component fit over a single-component. Only molecules that had greater than 75 binding events were analyzed.

Global fitting was performed on the step size distribution data to determine the parameters for a two-component Gaussian model. In this weighted global fit, data from each OM concentration contributed equally to the parameter estimation, and

data from each molecule contributed equally within a given OM concentration. Unweighted fitting produced similar results to this weighted approach. In the global fit, the means and standard deviations were shared between all datasets, but the relative amplitude of the two components could vary. Using a double Gaussian distribution to fit each condition individually was not statistically justified over a single component, as measured by the log-likelihood ratio test. Simulations showed however that this would be expected even if two populations of events were present with the distributions obtained via global fitting. This is because the relative size of the standard deviations of the step size distributions (~7–8 nm) is large compared to the expected difference between the mean step sizes (~0 nm and ~5.5 nm), and even several thousand simulated event measurements were not enough for two components to be statistically justified.

Weighting of molecules and conditions for the event duration global fit was performed as described above for the step size global fit.

**Fitted parameter calculations and reported uncertainties**. Calculations of the effective concentration leading to 50% of the observed effect (EC$_{50}$) were performed in Prism v 7.03 (GraphPad Software). Three parameters were used in the model (Min, Max, EC$_{50}$), and the standard error of the mean was reported for EC$_{50}$. Uncertainties in step size measurements are reported as standard errors of the mean, calculated from the step size distributions. Uncertainties for rates, proportions of events, and distance parameters are given as 95% confidence intervals calculated via 200–500 rounds of bootstrapping[24], corresponding to approximately twice the expected standard error of the mean.

**Stopped-flow experiments**. Transient kinetic experiments were performed with pyrene-labeled actin filaments using a stopped-flow instrument (Applied Photophysics)[47]. Porcine cardiac myosin was prepared from cryoground pig heart ventricles (PelFreeze)[48] and was then digested by chymotrypsin to isolate the S1 fragment[49]. Experiments were performed using the same assay buffer (AB) as used in the trapping experiments at room temperature (20 °C). All solutions contained a total of 0.1% DMSO. For AM·ADP dissociation experiments, 1 μM myosin S1 was incubated with 2 mM MgADP and 1 μM pyrene-labeled F-actin before being rapidly mixed with a solution containing 76 μM unlabeled F-actin, 2 mM MgADP, and 0.02 U mL$^{-1}$ apyrase. The pyrene fluorescence transient was fit to a single-exponential function plus a linear function (to account for drift over the long acquisition time). The rigor myosin dissociation experiments were conducted in a similar manner but without MgADP and apyrase.

**Simulations**. A Monte Carlo simulation using a modified Gillespie method[50] was utilized to relate the single-molecule biochemical and mechanical measurements to previous results from in vitro motility assays[9,10] and permeabilized cardiomyocyte assays[11,14]. To keep the model simple, we only included two states of the myosin, bound and unbound to actin, and ATP hydrolysis was ignored. A single thin filament from half a sarcomere was modeled, where 75 myosin molecules spaced 14.3 nm apart (coming from three thick filaments) could interact continuously with the infinitely stiff thin filament. Cooperative activation arising from strong binding of myosin heads to the thin filament was modeled by allowing heads near an already bound myosin head to exhibit the maximal on rate ($\varepsilon_{max}$·f), with an exponential decrease of the on rate as distance from the bound head increased[30]. Because the kinetics of OM binding to myosin has not been studied, we did not allow the exchange of OM with the myosin heads, but set the proportion of myosin with OM bound based the simulated concentration of OM and using a $K_D$ of 100 nM for motility simulations and 1.2 μM for cardiomyocyte simulations. Measured phosphate release rates from solution were used for the attachment rates to an undecorated actin filament. Model parameters were taken from the literature when possible, with only one parameter ($\varepsilon_{min}$) being adjusted to fit experimental control data (Supplementary Table 5). At each concentration of OM and calcium, 1000 steps of the simulation were run 1000 times. For the cardiomyocyte simulations, the actin position was held constant (to simulate isometric conditions) and the average force produced during the last half of the simulations was recorded. To simulate motility assay results, the simulations were run without any actin regulation and the total distance the actin filament traveled divided by the simulation time was used as the measured velocity. For the OM titration curves (Fig. 5c, Supplementary Figure 8c), the simulations displayed were run at pCa 6.4 since this was when the simulated force was ~15% activated at zero OM, as it was for pCa 6.0 in Nagy et al.[11] Forces for Fig. 5c, d and Supplementary Figure 8c, d were scaled to match the data from Nagy et al. by first subtracting the simulated force at 0 OM at pCa 9 (simulated passive force), then multiplying by a constant factor set to scale the force produced at pCa 4.75 (with no OM) in the simulations to that measured by Nagy et al. under the same conditions. This same constant factor (0.306) was used across all parameters, OM concentrations, and pCA values.

## Data availability
The data that support the findings of this study are available from the corresponding author upon reasonable request.

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

## Acknowledgements

This work was supported by the Center for Engineering MechanoBiology NSF Science and Technology Center, CMMI: 15-48571, National Institutes of Health grant R35-GM118139 to Y.E.G., P01-GM087253 to Y.E.G. and E.M.O., R01-HL133863 to D.A.W. and E.M.O., R00 HL123623 to M.J.G. and a National Science Foundation Graduate Research Fellowship to M.S.W.

## Author contributions

M.S.W. performed and analyzed optical trap experiments, stopped-flow experiments, and simulations. M.J.G. performed optical trap experiments and made the initial observations. B.B. and D.A.W. produced proteins and performed and analyzed motility assays.

E.M.O. and Y.E.G. directed research and contributed to analysis. M.S.W., E.M.O., and Y.E.G. wrote the manuscript. All authors reviewed and edited the manuscript.

## Additional information

**Competing interests:** The authors declare no competing interests.

