## [Peer Review File · Nature Communications]

Reviewers' comments:

Reviewer #1 (Remarks to the Author):

Omecamtiv mecarbil (OM) is a drug in phase III clinical trials for treatment of heart failure. It increases cardiac output, but the mechanism by which this occurs is unknown. Woody et al. perform a series of single molecule experiments and, on the basis of their results, argue that OM causes myosin to bind to actin (1) without performing a powerstroke, (2) in a force-independent manner, and (3) in an ATP-independent manner. Although these results might suggest that OM would decrease cardiac output, they then show, using a mathematical model, that the observed increases in force output from OM can arise from increases in cooperative activation of the thin filament. This mechanism can explain several apparently contradictory experimental results.

I am not an expert on omecamtiv mecarbil, and am largely unfamiliar with that literature, so I will not comment on the importance of this work on the treatment of heart disease or the understanding of that drug's action. However, I am familiar with single molecule experiments on myosin. I find the presented experimental work to be convincing. I have a few small comments, appended below.

I was specifically asked to comment on the modeling and simulations of muscle mechanics. My understanding is that this relates primarily to Fig. 5 of the main text, Supplementary table 5, and the final section of the Methods. Broadly, I also find these results convincing. As with the experimental work, I have a few small comments that I append below.

Overall, my sense is that this is a thorough study that presents a convincing and clear picture for the action of OM. I support its publication.

Minor comments:

(1) I was surprised to see that event lifetime in the absence of OM is described by a single exponential at all ATP concentrations. Specifically, I'd expect that detachment would occur via two sequential steps: i. ADP release (at $\sim 50\text{s}^{-1}$) followed by ii. ATP binding (at $\sim 3\ \mu\text{M}^{-1}\text{s}^{-1}$). Thus, event duration should be the sum of two exponentials, something like $T \sim ((50 \cdot 3 \cdot \text{ATP}) / (50 - 3 \cdot \text{ATP})) \cdot (\exp(-3 \cdot \text{ATP} \cdot t) - \exp(-50t))$ -- see, e.g. Rief et al. 2000, and their analysis of myosin V step duration. When one or the other is much faster than the other (e.g. high or low ATP), I would not be surprised that the data would be well-fit by a single exponential; however, when $\text{ATP} \sim 10\ \mu\text{M}$, the two rates are similar and I'd expect the data to require a double exponential fit. It's a little hard to evaluate the goodness of fit in Fig. 3a and Supplemental Fig. 5e, because of the scale on the horizontal axis. How was it determined that the data required a double vs. single exponential fit? I didn't see reference to a statistical test, but this seems well-suited for an F-test. This is a small detail, and not worth putting in the main text, but perhaps could be addressed in a supplemental note.

(2) In Supplemental Table 5, the cooperative distance for thin filament activation is given as $L = 157\text{nm}$. This is not consistent with the reference given, Longyear et al. 2017, who report $L = 400\text{nm}$.

I anticipate that this difference arises from scaling this distance by the spacing between myosin reported in Mijailovic et al. of 14.3nm with that assumed in Longyear et al. of 36nm : $(400\text{nm}) \cdot 14.3/36 = 159\text{nm}$. The estimate of Longyear et al. comes from measurements at the molecular level, combined with estimates of spacing between myosin molecules adsorbed to a surface (Harris and Warshaw 1993). Therefore, I am not sure that it makes sense to scale it in this way.

Another potentially confounding effect is that there are two protofilaments on actin, each with a tropomyosin "filament". Thus, it is possible that, while the spacing between myosin molecules is 14.3nm in a sarcomere, neighboring myosin molecules interact with different tropomyosin molecules and thus are not coupled. Thus, the effective spacing is 28.6nm .

These are rather esoteric details. Nevertheless, the choice of the parameter L should be discussed a little bit, particularly because it does not directly follow from the reference cited. I'd recommend another short supplementary note discussing some of these points.

(3) I really like Figure 5, but fear that it is too complex. I'd recommend moving all but the SEPTA

model fits to the supplement. Fig 5e is particularly hard to understand at first glance.

Reviewer #2 (Remarks to the Author):

The manuscript by Woody et al is an extremely provocative and thorough examination of the effects of Omecamtiv (OM) on human cardiac myosin at a single molecule level. The manuscript is beautifully presented and is accompanied by simulations that predict the effect of the drug on the ability of this myosin to propel actin in an in vitro motility assay, the contraction of muscle fibers and in the effect of the drug on the calcium sensitivity of fibers. This is all very important since the abundant literature on the effect of this drug is very confusing. Although it was first described as an activator of cardiac myosin, subsequent studies showed that it inhibited the rate of actin filament sliding in in vitro assays. The current study convincingly shows that OM eliminates the myosin powerstroke in a force independent manner giving rise to a prolonged attachment time on actin filaments. This subsequently affects the calcium sensitivity in simulations by altering the position of the TN-TM complex on actin. In my opinion this manuscript resolves most of the conflicting data that exists in the literature concerning the mechanism of action of the drug. I have only a few minor concerns that should be addressed.

Fig. 5b and text on the bottom of P.11: 4 lines from the bottom: I don't see a brown curve corresponding to the 10X increase in "d".

Same paragraph. The last sentence (which comes after the description of the brown curve) says that "increasing d 10-fold effectively reduces the mean detachment rate to $\sim 10/s$..." Do you actually mean decreasing k_{detach} here?

Last two sentences of the Results. It is mentioned that a dissociation constant for OM of 1.2 μM was used in the simulations in contrast with the 100 nM value determined from the optical tapping. This should be better justified. Studies with blebbistatin clearly show that more drug is required to inhibit actin gliding than to inhibit the steady state actin-activated ATPase activity and this is probably due to the nature of the two assays. In the ATPase assay you are simply measuring the average value of the inhibited and uninhibited myosins, whereas in the motility assay you have a more complex situation involving noninhibited myosins acting on actin with the inhibited myosin behaving benignly or perhaps only slightly inhibitory due to molecular friction. Since this assay is largely independent of the number of cycling heads you might not expect significant inhibition of the observed sliding velocity until a large fraction of the motors are inhibited. This is due to the fact that blebbistatin inhibition leaves myosin in a weak actin-binding state. In contrast, if OM is essentially acting to force myosin into a strongly bound state and if this is independent of the direction of applied force, you might expect to see it being dominant in the in vitro motility assay as noted by Swenson et al, yet your results are against this. Have you tried to simulate these data? The affinities and EC50 are discussed in supplemental materials, but I think it would be helpful if some of this were moved to the main text if space is available. Could you please elaborate?

Reviewer #3 (Remarks to the Author):

Prof. Woody and colleagues present the results of an enquiry into mechanism of action of the cardiac myosin activator, omecamtiv mecarbil (OM). The group is well known for their work and the experimental approach is elegant, although there are some limitations that would best be addressed.

General Comments

1) Impact of Dose Dependence on Results and Conclusions:

The authors are to be commended for performing a dose response at the start of the paper. However, throughout the paper, the authors emphasise the effects at 10 μM which is 100x higher than clinically relevant concentrations of OM. The authors should avoid generalising the findings at 10 μM OM, as occurs in the lead off sentence and throughout the manuscript. Several experiments are done (e.g. Fig 3e) using this 10 μM OM concentration. The authors should note early in the paper that the clinically relevant concentrations of OM range from 200 ng/mL to 600 ng/mL in plasma and the free fraction is about 20% (Vu, T., Ma, P., Xiao, J. J., Wang, Y.-M. C., Malik, F. I., & Chow, A. T. (2015). Population pharmacokinetic-pharmacodynamic modeling of omecamtiv mecarbil, a cardiac myosin activator, in healthy volunteers and patients with stable heart failure.

Journal of Clinical Pharmacology, 55(11), 1236–1247.), leading to clinically relevant free plasma concentrations ranging from 100 – 300 nM. The authors should repeat the experiments done only at a single concentration of 10 μ M at a more relevant OM concentration of say 200 nM.

2) Impact of Nucleotide State of Cardiac Myosin on Results and Conclusions:

The results of Planelles-Herrero, et al (Ref 30) suggest the affinity of OM for myosin varies substantially dependent on the nucleotide state of cardiac myosin. The use of high concentrations of OM may bias the findings of this paper since it forces occupancy of a state not otherwise populated at clinically relevant OM concentrations. How do the results in this paper account for this issue? Is it possible that OM dissociates from myosin after myosin binds to the actin filament thus minimising the effect on progression through the rest of the powerstroke?

3) Impact of Temperature Dependence of Results and Conclusions:

Myosin is a highly temperature dependent enzyme – the temperature of the optical trap assay is not included in the methods (please provide this information). Assuming these results were generated at room temperature, how would these results vary if the temperature were at a physiologic 37C? Recognising that it may not be practical to conduct these experiments at normal body temperature, the authors should at least discuss the implications of the lower than physiological temperature of their assay preparation on the kinetic rate constants of the individual steps in the cycle and the potential impact to their model and conclusions.

4) Clinical relevance:

Given the aforementioned concerns, it is unclear how these findings provide useful insight into the clinical utility or function of OM.

5) Stylistic:

In general, the authors should avoid words with emotional impact like “dramatic”.

Specific Comments: Methods and Results

1) The authors went to some trouble to express and use human cardiac HMM myosin. However skeletal muscle actin rather than cardiac muscle actin is used for all the in vitro assays – are there any differences if cardiac actin is used?

2) Model – the hypothesis that OM completely inhibits the working stroke seems to be derived from findings made at 10 μ M OM. This concentration could be the one where you get complete occupancy of myosin by OM and so the events are the only ones you observe. However, as noted above, this concentration could also result in OM binding to states whose affinity for OM is quite low and thus potentially irrelevant at clinically relevant exposures. In this context – the following statement seems too definitive.

This result is consistent with a model in which myosin performs its full working stroke when OM is not bound but has virtually no net displacement when OM is bound, and the proportion of interactions that occur with OM is bound depends on the OM concentration.

3) It is not clear how the model accounts for the non-steady state conditions of cardiac contraction which generally lasts for only 300 ms, or only enough time for roughly 1-2 turnovers of myosin before systole is complete. Isometric conditions are never achieved. An alternate model could be that more myosins in the pre-powerstroke state prior to the onset of systole lead to greater activation of the thin filament. A modest slowing of the powerstroke could prolong thin filament activation or merely the presence of more heads on the thin filament could do so – how do these experiments distinguish between the two?

4) The stipulation of a non-canonical pathway for detachment is dependent on measurements made at 10 μ M OM and varying ATP concentration. The strong dose dependence of effect on the step size (and progression through the mechanochemical cycle) warrant an experiment at a lower OM concentration (e.g. 200 nM). The conclusion is otherwise misleading given it is not made based on clinically relevant conditions.

5) The SEPTA model seems to be a steady-state model – its applicability to the cardiac cycle seems limited given the cardiac sarcomere doesn't remain activated long enough for multiple turnovers to occur. Yet the authors extrapolate it to explain cardiac physiology later on. Is the model relevant to the beating heart?

6) The justification of switching the dissociation constant of OM from 100 nM to 1.2 μ M in myocytes seems to be a bit artificial. Malik et al (Science 2011, Supplemental Online Material) reported that cardiac myocytes had a maximal increase in contractility (Fractional Shortening) at 400 nM. At 800 nM, the cardiac myocytes shrank substantially, and their contractility was no longer increased. Thus the selection of 1.2 μ M seems excessive.

7) OM increases cardiac contractility in the absence of changes in the velocity of contraction or relaxation at clinically relevant concentrations (no changes in +/- dP/dt) in an intact dog model of HF (Malik, Science 2011, Shen Circ HF, 2011). How do these in vitro findings correlate with the findings in intact preclinical (and human) models of cardiac function?

Specific Comments: Discussion

1) The conclusion that OM is not a direct activator of myosin seems to be a matter of semantics. OM places myosin in a favourable conformational state to bind to actin at the start of the cardiac cycle and leads to more heads binding to the thin filament – this seems to be an activating mode of action. The authors should consider restatement.

2) The statement that it eliminates the working stroke also are dependent on observations at high OM concentrations. It is very possible that OM dissociates from myosin after actin binding and then myosin progresses through its cycle normally. How do these experiments account for that possibility?

3) The FRET experiments cited in the discussion were similarly conducted at very high OM concentrations. The same caveats apply as in the current study.

4) The authors do not comment on how their proposed mechanism of action seems to be a more efficient way to increase contractility than to merely accelerate ATP turnover. The energetic benefits of this mechanism of action should be discussed.

5) Arrhythmias have not been observed in the clinical program for OM, even at high drug exposures. In fact, in the acute heart failure study (ATOMIC-AHF) there was a suggestion of reduced atrial arrhythmias. The authors should delete this statement from the discussion.

6) Selective inhibitors of myosin have been developed and are being pursued clinically for their effect to decrease cardiac contractility. The nomenclature the authors propose to use in this paper does not distinguish between "inhibitors" that increase contractility and "inhibitors" that decrease contractility. The authors' use of the word inhibitor has the potential to be very confusing clinically.

We sincerely appreciate the Reviewers' thoughtful comments, as their input has resulted in a stronger paper. Below, the original Review comments are in black, and our responses are red. Changes in the revised manuscript also appear in red.

Reviewers' comments:

Reviewer #1 (Remarks to the Author):

Omecamtiv mecarbil (OM) is a drug in phase III clinical trials for treatment of heart failure. It increases cardiac output, but the mechanism by which this occurs is unknown. Woody et al. perform a series of single molecule experiments and, on the basis of their results, argue that OM causes myosin to bind to actin (1) without performing a powerstroke, (2) in a force-independent manner, and (3) in an ATP-independent manner. Although these results might suggest that OM would decrease cardiac output, they then show, using a mathematical model, that the observed increases in force output from OM can arise from increases in cooperative activation of the thin filament. This mechanism can explain several apparently contradictory experimental results.

I am not an expert on omecamtiv mecarbil, and am largely unfamiliar with that literature, so I will not comment on the importance of this work on the treatment of heart disease or the understanding of that drug's action. However, I am familiar with single molecule experiments on myosin. I find the presented experimental work to be convincing. I have a few small comments, appended below.

We thank the reviewer for their appreciation of the experimental work.

I was specifically asked to comment on the modeling and simulations of muscle mechanics. My understanding is that this relates primarily to Fig. 5 of the main text, Supplementary table 5, and the final section of the Methods. Broadly, I also find these results convincing. As with the experimental work, I have a few small comments that I append below.

We thank the reviewer for the positive comments regarding the simulations.

Overall, my sense is that this is a thorough study that presents a convincing and clear picture for the action of OM. I support its publication.

Minor comments:

(1) I was surprised to see that event lifetime in the absence of OM is described by a single exponential at all ATP concentrations. Specifically, I'd expect that detachment would occur via two sequential steps: i. ADP release (at $\sim 50\text{s}^{-1}$) followed by ii. ATP binding (at $\sim 3\ \mu\text{M}^{-1}\text{s}^{-1}$). Thus, event duration should be the sum of two exponentials, something like $T \sim ((50 + 3 \cdot \text{ATP}) / (50 \cdot 3 \cdot \text{ATP})) \cdot (\exp(-3 \cdot \text{ATP} \cdot t) - \exp(-50t))$ -- see, e.g. Rief et al. 2000, and their analysis of myosin V step duration. When one or the other is much faster than the other (e.g. high or low ATP), I would not be surprised that the data would be well-fit by a single exponential; however, when

ATP~10 μM , the two rates are similar and I'd expect the data to require a double exponential fit. It's a little hard to evaluate the goodness of fit in Fig. 3a and Supplemental Fig. 5e, because of the scale on the horizontal axis. How was it determined that the data required a double vs. single exponential fit? I didn't see reference to a statistical test, but this seems well-suited for an F-test. This is a small detail, and not worth putting in the main text, but perhaps could be addressed in a supplemental note.

The reviewer is correct that with sufficient time resolution, the duration of attachment would likely be better described by a double exponential at ATP concentrations near 10 μM . However, because of the 20 ms deadtime that is present in this data due to the smoothing of the covariance signal for event detection, the "lag-phase" of the double exponential arising from the 50 s^{-1} rate of ADP release is not fully observed and thus the available data is best fit by a single exponential. This is what we expect from simulations we have performed given these rates and deadtime. We have added the following text to the first supplemental note to explain this:

For example, in our data at 10 μM MgATP with no OM, we might expect the distribution of durations to be described the sum of two exponentials since the ADP release rate ($\sim 50 \text{ s}^{-1}$) is close the ATP binding rate ($\sim 30 \text{ s}^{-1}$). However, because of the 20 ms deadtime for detecting events from the bead covariance, the lag-phase expected from the $\sim 50 \text{ s}^{-1}$ ADP release rate is not observed, and the data are well fitted by a single exponential corresponding to the ATP binding rate.

We used the log-likelihood ratio test to determine the significance of fitting to a double exponential distribution. We have made this clearer by referencing the test in the text (page 6) as well as in the methods (page 23). We also added the p-values for this test in the caption for Supplementary Figure 5 and in Supplementary Table 3. We also added a note to Supplementary Table 3 explaining a small detail regarding the fitting of data at 4 mM MgATP with no OM.

(2) In Supplemental Table 5, the cooperative distance for thin filament activation is given as $L=157\text{nm}$. This is not consistent with the reference given, Longyear et al. 2017, who report $L=400\text{nm}$.

I anticipate that this difference arises from scaling this distance by the spacing between myosin reported in Mijailovic et al. of 14.3nm with that assumed in Longyear et al. of 36nm:

$(400\text{nm}) \cdot 14.3/36 = 159\text{nm}$. The estimate of Longyear et al. comes from measurements at the molecular level, combined with estimates of spacing between myosin molecules adsorbed to a surface (Harris and Warshaw 1993). Therefore, I am not sure that it makes sense to scale it in this way.

Another potentially confounding effect is that there are two protofilaments on actin, each with a tropomyosin "filament". Thus, it is possible that, while the spacing between myosin molecules is 14.3nm in a sarcomere, neighboring myosin molecules interact with different tropomyosin molecules and thus are not coupled. Thus, the effective spacing is 28.6 nm.

These are rather esoteric details. Nevertheless, the choice of the parameter L should be discussed a little bit, particularly because it does not directly follow from the reference cited. I'd recommend another short supplementary note discussing some of these points.

We appreciate the attention regarding this important parameter and agree that the presentation of the parameters was not ideal. We chose $L = 157$ nm to obtain a coupling constant, $C=L/\Delta s$, (where Δs is the myosin spacing) equal to 11 as used by Longyear et al. for motility assay simulations and cellular-scale simulations. In fact, because our actin/thin filaments are infinitely stiff in our simulations, the spacing between myosin heads (Δs , which we set at 14.3 nm) has no direct effects on the motility or force results. To clarify this, we have updated Supplementary Table 5 to reflect C instead of L and removed the parameter, Δs . We also added the following footnote to the table explaining C :

**This constant was used for simulations of motility assays and cellular-scale simulations as in Longyear et. al., (2017) although it may not directly correspond to the cooperative unit in a native thin filament.

(3) I really like Figure 5, but fear that it is too complex. I'd recommend moving all but the SEPTA model fits to the supplement. Fig 5e is particularly hard to understand at first glance.

We appreciate the reviewer's praise for Figure 5 and agree it was complex. We have moved the existing Figure 5 to become Supplemental Figure 8 and have replaced it in the main text with a simplified version with only the Malik and SEPTA Models, except for Panel e which only has the SEPTA model. We think comparing the SEPTA model to the Malik model in panels B and C is important to demonstrate to non-experts that the observed effects cannot be explained by the originally proposed mechanism. We trust that this simpler format is much easier to understand.

Reviewer #2 (Remarks to the Author):

The manuscript by Woody et al is an extremely provocative and thorough examination of the effects of Omecamtiv (OM) on human cardiac myosin at a single molecule level. The manuscript is beautifully presented and is accompanied by simulations that predict the effect of the drug on the ability of this myosin to propel actin in an in vitro motility assay, the contraction of muscle fibers and in the effect of the drug on the calcium sensitivity of fibers. This is all very important since the abundant literature on the effect of this drug is very confusing. Although it was first described as an activator of cardiac myosin, subsequent studies showed that it inhibited the rate of actin filament sliding in in vitro assays.

The current study convincingly shows that OM eliminates the myosin powerstroke in a force independent manner giving rise to a prolonged attachment time on actin filaments. This subsequently affects the calcium sensitivity in simulations by altering the position of the TN-TM complex on actin. In my opinion this manuscript resolves most of the conflicting data that exists in the literature concerning the mechanism of action of the drug.

We thank the reviewer for his/her kind words.

I have only a few minor concerns that should be addressed.

Fig. 5b and text on the bottom of P.11: 4 lines from the bottom: I don't see a brown curve corresponding to the 10X increase in "d".

Same paragraph. The last sentence (which comes after the description of the brown curve) says that "increasing d 10-fold effectively reduces the mean detachment rate to $\sim 10/s$..." Do you actually mean decreasing k_{detach} here?

Upon consideration of this comment and the last comment of reviewer 1, we have simplified Figure 5 in the main text by including only the Malik and SEPTA models there, while moving the former Figure 5, containing all model results, to Supplemental Figure 8. We also have slightly changed the parameters for the SE and PTA models so that k_0 and d are altered together in the PTA model and only the step size (and not d) is changed in the SE model. This is a more cogent comparison than our previous set of model parameters. As a result of these changes, the brown line is now more apparent in Supplemental Figure 8. We also have changed the notation from k_{detach} to $k_{detach0}$ in Figure 5a and Supplemental Figure 8a to reflect that this number is the detachment rate at zero load, and the actual detachment rate is calculated from the Bell equation using the unloaded rate ($k_{detach0}$) and the distance parameter, d .

Last two sentences of the Results. It is mentioned that a dissociation constant for OM of 1.2 μM was used in the simulations in contrast with the 100 nM value determined from the optical tapping. This should be better justified. Studies with blebbistatin clearly show that more drug is required to inhibit actin gliding than to inhibit the steady state actin-activated ATPase activity and this is probably due to the nature of the two assays. In the ATPase assay you are simply measuring the average value of the inhibited and uninhibited myosins, whereas in the motility assay you have a more complex situation involving noninhibited myosins acting on actin with the inhibited myosin behaving benignly or perhaps only slightly inhibitory due to molecular friction. Since this assay is largely independent of the number of cycling heads you might not expect significant inhibition of the observed sliding velocity until a large fraction of the motors are inhibited. This is due to the fact that blebbistatin inhibition leaves myosin in a weak actin-binding state. In contrast, if OM is essentially acting to force myosin into a strongly bound state and if this is independent of the direction of applied force, you might expect to see it being dominant in the in vitro motility assay as noted by Swenson et al, yet your results are against this. Have you tried to simulate these data? The affinities and EC50 are discussed in supplemental materials, but I think it would be helpful if some of this were moved to the main text if space is available. Could you please elaborate?

In the last two sentences on p. 12, we have stated that only the cardiomyocyte isometric force simulations were performed with a K_D for OM of 1.2 μM , while the motility assay simulations used $K_d = 100$ nM. This statement may not have been clear originally, so we have reworded the sentences at the end of the results on page 12 as follows:

While the in vitro motility assay simulations used a dissociation constant for OM (K_d) of 100 nM, which is consistent with our optical trapping data, in the simulations of myocyte isometric force, K_d for OM was set to 1.2 μM . This value is consistent with the observed effect of the drug in the same type of muscle preparations (permeabilized rat ventricular trabeculae) used in Nagy et al. (2015) and as discussed in the Supplementary Note 3.

Because the effect on motility occurs very close to the K_d of 100 nM used in the motility simulations, it appears that the effect of OM inducing strong binding dominates the motility assay, consistent with our model and with Swenson et al.

Reviewer #3 (Remarks to the Author):

Prof. Woody and colleagues present the results of an enquiry into mechanism of action of the cardiac myosin activator, omecamtiv mecarbil (OM). The group is well known for their work and the experimental approach is elegant, although there are some limitations that would best be addressed.

General Comments

1) Impact of Dose Dependence on Results and Conclusions:

The authors are to be commended for performing a dose response at the start of the paper. However, throughout the paper, the authors emphasise the effects at 10 μM which is 100x higher than clinically relevant concentrations of OM. The authors should avoid generalising the findings at 10 μM OM, as occurs in the lead off sentence and throughout the manuscript. Several experiments are done (e.g. Fig 3e) using this 10 μM OM concentration. The authors should note early in the paper that the clinically relevant concentrations of OM range from 200 ng/mL to 600 ng/mL in plasma and the free fraction is about 20% (Vu, T., Ma, P., Xiao, J. J., Wang, Y.-M. C., Malik, F. I., & Chow, A. T. (2015). Population pharmacokinetic-pharmacodynamic modeling of omecamtiv mecarbil, a cardiac myosin activator, in healthy volunteers and patients with stable heart failure. *Journal of Clinical Pharmacology*, 55(11), 1236–1247.), leading to clinically relevant free plasma concentrations ranging from 100 – 300 nM. The authors should repeat the experiments done only at a single concentration of 10 μM at a more relevant OM concentration of say 200 nM.

We agree that the concentration of OM is of vital importance for this study, which is why all key experiments were performed over concentrations ranging from 50 nM to 10 μM . The only experiments for which only 10 μM OM was reported are those presented in Figure 3e at lower than physiological ATP concentrations and force dependence of detachment (Figure 4). The step size and attachment durations data at saturating ATP (the physiologically relevant [ATP]) show a dose dependent effect of OM with an EC50 (~100 nM) less than the plasma concentration range given by the reviewer.

The force dependent measurements of Fig 4 were done at 10 μM OM to allow analysis of this complex experiment without having to account for mixed populations of OM-bound and unbound actomyosin interactions. As mentioned in the paper's discussion (paragraph 5), there are compelling reasons to expect this force independent attachment to occur when the stroke is inhibited, which was shown to occur at a range of concentrations (Figure 2).

The experiment of 3e was performed to help determine further details about the mechanism of detachment of myosin when OM is bound, and thus a high concentration of 10 μM was necessary to ensure the drug was bound in the majority of interactions. The results of this experiment do not impact our proposed model for how the drug works *in vivo*, since the low ATP conditions of this experiment are not physiological. But they help elucidate OM's mechanism. Although the high OM concentration used in Figure 3e was appropriate for drawing the conclusion that OM causes dissociation independent of ATP-binding, we have performed an additional key supportive experiment at 200 nM ATP and 1000 nM OM (closer to the therapeutic range of the drug, but still high enough to nearly saturate binding, given the EC50 = 100 nM). We found nearly the same results as that presented in Figure 3e: $k_a = 0.875 \text{ s}^{-1}$ and a faster k_b of 7.57 s^{-1} , supporting all of our presented conclusions. A figure of this data is shown below to

address the reviewer's concerns, but it would be distracting to include this in the paper or supplement.

We have added the following sentence on page 5 when the concentration dependent effects of OM are first discussed, making reference to the Vu et al study:

This effect occurs near the clinically relevant plasma concentration of OM of 100-600 nM²³.

This additional experimental data and the reasoning given above gives compelling evidence that the drug-induced ATP-independent dissociation is present at therapeutically relevant drug concentrations.

2) Impact of Nucleotide State of Cardiac Myosin on Results and Conclusions:

The results of Planelles-Herrero, et al (Ref 30) suggest the affinity of OM for myosin varies substantially dependent on the nucleotide state of cardiac myosin. The use of high concentrations of OM may bias the findings of this paper since it forces occupancy of a state not otherwise populated at clinically relevant OM concentrations. How do the results in this paper account for this issue? Is it possible that OM dissociates from myosin after myosin binds to the actin filament thus minimising the effect on progression through the rest of the powerstroke?

As discussed above, nearly all experiments were performed with 50 nM to 10 μ M OM. Notably, the step size and attachment duration measurements at saturating ATP concentrations clearly show the effects of OM at clinically relevant concentrations. Planelles-Herrer et al. report

affinities $\geq 1.8 \mu\text{M}$ for all tested biochemical states besides the ADP-VO₄ state ($K_d \sim 300 \text{ nM}$), and the effects on step size and attachment durations we found are clearly detected at and below this concentration. We do acknowledge the data in the Planelles-Herrero et al. paper in the Discussion and supplemental notes (Ref 32) and consider that it supports our conclusion that the pre-powerstroke state is stabilized. In answer to the second question, we observe the entire interaction between myosin and actin and thus, if OM dissociated from myosin and the power stroke then occurred, it would be resolved directly in the experiments. This is not the observed result, however, as reported in Figure 2.

3) Impact of Temperature Dependence of Results and Conclusions:

Myosin is a highly temperature dependent enzyme – the temperature of the optical trap assay is not included in the methods (please provide this information). Assuming these results were generated at room temperature, how would these results vary if the temperature were at a physiologic 37C? Recognising that it may not be practical to conduct these experiments at normal body temperature, the authors should at least discuss the implications of the lower than physiological temperature of their assay preparation on the kinetic rate constants of the individual steps in the cycle and the potential impact to their model and conclusions.

We apologize for not including the temperature information. It has been added to the methods section under “Optical Trapping Assays”. The experiments were performed at $20 \pm 1^\circ\text{C}$. This is consistent with the biochemical work initially reported for the drug (Malik et al., Science, 2011) which was performed at 25°C , as well as other biochemical studies of the drug’s effects (Liu et al, Biochemistry, 2015). While it is known that increased temperature acts to increase the rate of many biochemical parameters of myosin, the general behavior of myosin and the enzymatic reaction path remain the same at higher temperature (Tombe, P. P. D. & Stienen, G. J. M. Impact of temperature on cross-bridge cycling kinetics in rat myocardium. The Journal of Physiology 584, 591–600). The kinetics of detachment are likely to be increased at physiological temperature, but this work does not address muscle tension development rate. Thus discussion of the temperature dependence of our results without any relevant data is beyond the scope of the paper.

4) Clinical relevance:

Given the aforementioned concerns, it is unclear how these findings provide useful insight into the clinical utility or function of OM.

In order to understand the therapeutic action of the drug, its interaction with other pharmaceuticals, its therapeutic window, toxicity at high doses, and to assist discovery of other cardiotoxic agents, the biophysical mechanism of the drug is essential. Since all of the experiments which led to our updated mechanism were done under physiologically or clinically relevant ATP and OM concentrations, our results are directly relevant to the clinical utility and function of OM. The fundamental mechanism of drug action requires study over a range of concentrations including below and above the therapeutic window. The present study provides novel and important results for better understanding the drug’s action at all concentrations.

5) Stylistic:

In general, the authors should avoid words with emotional impact like “dramatic”.

We have removed the word dramatic and instead substituted “the 30-fold reduction in gliding velocity..” on page 13.

Specific Comments: Methods and Results

1) The authors went to some trouble to express and use human cardiac HMM myosin. However skeletal muscle actin rather than cardiac muscle actin is used for all the in vitro assays – are there any differences if cardiac actin is used?

Although very slight differences in the effect of OM in the presence of cardiac vs skeletal thin filaments have been observed (Malik et. al, 2011), cardiac and skeletal actin differ by only 4 amino acids, and previous work has not shown any difference in biochemical or motility parameters between skeletal, smooth muscle, and cardiac actin isoforms interacting with a variety of myosins, including beta cardiac myosin (Harris and Warshaw, *Circulation Research*, 1993, Bookwalter and Trybus, *JBC*, 2006). These findings have led to the field generally accepting studies using skeletal actin for studying a wide variety of myosins.

2) Model – the hypothesis that OM completely inhibits the working stroke seems to be derived from findings made at 10 μ M OM. This concentration could be the one where you get complete occupancy of myosin by OM and so the events are the only ones you observe. However, as noted above, this concentration could also result in OM binding to states whose affinity for OM is quite low and thus potentially irrelevant at clinically relevant exposures. In this context – the following statement seems too definitive.

This result is consistent with a model in which myosin performs its full working stroke when OM is not bound but has virtually no net displacement when OM is bound, and the proportion of interactions that occur with OM is bound depends on the OM concentration.

As mentioned above, the conclusion about the inhibition of the step size comes from data taken at OM concentrations of 50 nM, 100 nM, 200 nM, 500 nM, and 10 μ M over which range the proportion of zero-force and full force events gradually increases. These data show a dose dependent effect (Figure 2f), indicating that the step size is inhibited at low OM concentrations (\leq 500 nM). Thus, our statement is an appropriate expression of the data.

3) It is not clear how the model accounts for the non-steady state conditions of cardiac contraction which generally lasts for only 300 ms, or only enough time for roughly 1-2 turnovers of myosin before systole is complete. Isometric conditions are never achieved. An alternate model could be that more myosins in the pre-powerstroke state prior to the onset of systole lead to greater activation of the thin filament. A modest slowing of the powerstroke could prolong thin filament activation or merely the presence of more heads on the thin filament could do so – how do these experiments distinguish between the two?

Since accurately modeling the twitch (non-steady state contraction) of muscle is its own current area of active research, we have not attempted to model this phenomenon for this study. We instead have shown how our observations of an inhibited powerstroke are consistent with the observed effects in muscle fibers of increased isometric force previously observed by other investigators. Our experiments bear against a slower power stroke because we do not observe a slowing of the powerstroke, but rather an inhibition of the stroke. Thus, our data supports the

idea that more bound heads lead to thin filament activation.

4) The stipulation of a non-canonical pathway for detachment is dependent on measurements made at 10 μM OM and varying ATP concentration. The strong dose dependence of effect on the step size (and progression through the mechanochemical cycle) warrant an experiment at a lower OM concentration (e.g. 200 nM). The conclusion is otherwise misleading given it is not made based on clinically relevant conditions.

We responded to this point above related to the first comment of the reviewer. We do not claim that low ATP is physiologically relevant, but these conditions provide mechanistic insight into events that occur when OM is bound to myosin. The experiments at saturating ATP with the full range of OM concentrations are sufficient to draw the conclusions of the SEPTA model.

5) The SEPTA model seems to be a steady-state model – its applicability to the cardiac cycle seems limited given the cardiac sarcomere doesn't remain activated long enough for multiple turnovers to occur. Yet the authors extrapolate it to explain cardiac physiology later on. Is the model relevant to the beating heart?

We have generally answered this earlier related to Minor Point #3 above. More specifically, while the simulations we performed simulated steady-state experiments, the SEPTA effects of step-elimination and prolonged time of attachment are not specific to a steady state model, but are derived from experiments using actively cycling myosin.

6) The justification of switching the dissociation constant of OM from 100 nM to 1.2 μM in myocytes seems to be a bit artificial. Malik et al (Science 2011, Supplemental Online Material) reported that cardiac myocytes had a maximal increase in contractility (Fractional Shortening) at 400 nM. At 800 nM, the cardiac myocytes shrank substantially, and their contractility was no longer increased. Thus the selection of 1.2 μM seems excessive.

As explained in Supplementary Note 3 in the paper, the use of 1.2 μM for the K_d for the cardiomyocyte simulations was used because the permeabilized rat trabeculae used in the experiments from Nagy et al. showed an effective EC50 of 1.2 μM for ATPase inhibition. We acknowledge that the isolated rat cardiomyocytes from heart ventricles in the Malik et al. paper showed a different concentration dependence. Because of the variability of the concentration dependence between the different muscle preparations, we used a value determined from the system that more closely matches the one we simulated. We clarified this in the main text by changing the last sentence of the results to:

This value is consistent with the observed effect of the drug in the same type of muscle preparations (permeabilized rat trabeculae) used in Nagy et al. (2015) and as discussed in Supplementary Note 3.

7) OM increases cardiac contractility in the absence of changes in the velocity of contraction or relaxation at clinically relevant concentrations (no changes in $\pm dP/dt$) in an intact dog model of HF (Malik, Science 2011, Shen Circ HF, 2011). How do these in vitro findings correlate with the findings in intact preclinical (and human) models of cardiac function?

As stated earlier, complete modeling of the twitch from biochemical and mechanical parameters is still poorly understood that it is an area of active research. We don't claim provide complete explanations for all drug effects, but as appreciated by Reviewer 2, the explanations of several results which contradict the originally proposed mechanism are an advance for the field.

Specific Comments: Discussion

1) The conclusion that OM is not a direct activator of myosin seems to be a matter of semantics. OM places myosin in a favourable conformational state to bind to actin at the start of the cardiac cycle and leads to more heads binding to the thin filament – this seems to be an activating mode of action. The authors should consider restatement.

A 'direct activator' of myosin implies that it **directly** increases myosin function. Although overall cardiac function can be improved due to the thin filament activation, the direct function of myosin (generating force while attached to actin) is inhibited by the drug as shown by our step size measurements.

2) The statement that it eliminates the working stroke also are dependent on observations at high OM concentrations. It is very possible that OM dissociates from myosin after actin binding and then myosin progresses through its cycle normally. How do these experiments account for that possibility?

The same question was raised by the reviewer in General Comments #2. Please refer to the answer there. We again point out that the conclusions of inhibited step are supported by measurements performed at a range of OM concentrations.

3) The FRET experiments cited in the discussion were similarly conducted at very high OM concentrations. The same caveats apply as in the current study.

While this is true, our experiments were also performed at lower OM concentrations and provided consistent results. Thus it is relevant and important to cite this study.

4) The authors do not comment on how their proposed mechanism of action seems to be a more efficient way to increase contractility than to merely accelerate ATP turnover. The energetic benefits of this mechanism of action should be discussed.

It is outside the scope of the present work to extrapolate to the energetics of the mechanism of activation, as muscle energy usage is hard to predict. In our model, although OM bound myosins would use less ATP, the non-OM bound myosins that are recruited because of the thin filament activation might use more ATP. Without more information that is not available, conjecturing about the efficiency of contraction in the presence of OM is not warranted.

5) Arrhythmias have not been observed in the clinical program for OM, even at high drug exposures. In fact, in the acute heart failure study (ATOMIC-AHF) there was a suggestion of reduced atrial arrhythmias. The authors should delete this statement from the discussion.

We have removed this statement and replaced it with the following:

The concentration of OM in patients has been shown clinically to be crucial, as doses of OM above the therapeutic level slow contraction to the point where ischemia or other adverse reactions may occur³. [(Teerlink et al. 2011)]

6) Selective inhibitors of myosin have been developed and are being pursued clinically for their effect to decrease cardiac contractility. The nomenclature the authors propose to use in this paper does not distinguish between “inhibitors” that increase contractility and “inhibitors” that decrease contractility. The authors’ use of the word inhibitor has the potential to be very confusing clinically.

We have removed the last sentence of the conclusion which contained the word inhibitors and removed the word inhibit from the last sentence of the introduction.

REVIEWERS' COMMENTS:

Reviewer #1 (Remarks to the Author):

The authors have addressed my concerns.

Reviewer #2 (Remarks to the Author):

I am satisfied with the responses of the authors.